# SocREval: LLMs with the Socratic Method for Reference-free Reasoning Evaluation

## Abstract

To comprehensively assess the capacity of current models for complex reasoning, it is crucial to assess their step-by-step reasoning in a scalable manner. Established reference-based evaluation metrics rely on human-annotated reasoning chains to assess the model-derived chains. However, such "gold-standard" human-written reasoning chains may not be unique and their acquisition is often labor-intensive. Existing reference-free reasoning metrics eliminate the need for human-crafted reasoning chains as references, but they typically require fine-tuning on datasets with human-derived reasoning chains, which complicates the process and raises concerns regarding generalizability across diverse datasets. To address these challenges, we harness GPT-4 to automatically evaluate reasoning chain quality, obviating the need for human-crafted references. Leveraging the Socratic method, we devise tailored prompts to enhance reference-free reasoning evaluation, which we term SocREval (**Soc**ratic method for **R**easoning **Eval**uation). Empirical results from four human annotated datasets reveal that SocREval significantly improves GPT-4's performance, surpassing existing reference-free and reference-based reasoning evaluation metrics. Beyond its demonstrated efficacy, our proposed framework, large language models (LLMs) with the Socratic method, proves to be both cost-efficient and robust to prompt writing and example selection, as substantiated by our in-depth analysis.

## 1 Introduction

Recent advances in large language models (LLMs) have led to state-of-the-art results in a plethora of natural language processing (NLP) tasks, demonstrating the effectiveness of in-context learning without the need for task-specific training or fine-tuning (OpenAI, 2023; Anil et al., 2023; Touvron et al., 2023). Despite these impressive achievements, the inherent reasoning capabilities of LLMs remain notably below human expectations (Arkoudas, 2023). Although the core of reasoning fundamentally involves offering justifications, most contemporary evaluations primarily assess a model's reasoning capability based on its end-task performance (Huang & Chang, 2022). Such evaluations, focusing solely on the accuracy of the final answer, neglect the complexities of the underlying reasoning chains. This oversight inhibits a comprehensive understanding of a model's reasoning ability and poses challenges to further advancements in this domain.

To assess the quality of reasoning chains produced by models, a direct and intuitive approach centers on contrasting these generated chains with human-constructed ones, termed as reference-based reasoning evaluation (Clinciu et al., 2021; Welleck et al., 2022; Saparov & He, 2022). However, these reference-based metrics highly rely on human-constructed reasoning chains, which are both labor-intensive and costly. Furthermore, "gold-standard" reasoning chains may not be unique (Dalvi et al., 2021), implying that the effectiveness of reference-based evaluations can be significantly influenced by the choice and breadth of human-crafted references. In light of these challenges, recent research has begun to explore the evaluation of reasoning chains without necessitating human-annotated references—termed reference-free reasoning evaluation (Golovneva et al., 2022; Prasad et al., 2023). Regrettably, these reference-free metrics necessitate the fine-tuning of models on datasets with human-annotated reasoning chains, which is not only complicated but also restricts their applicability across diverse datasets.

---

**GSM8K**

**Question:** Janet's ducks lay 16 eggs per day. She eats three for breakfast every morning and bakes muffins for her friends every day with four. She sells the remainder at the farmers' market daily for $2 per fresh duck egg. How much in dollars does she make every day at the farmers' market?

**Generated response:** Janet eats 3 duck eggs for breakfast and bakes 4 into muffins so 3 + 4 = <<3+4=7>>7 duck eggs are used||Each day Janet's ducks lay 16 eggs and she uses 7, 16 - 7 = <<16-7=9>>9 duck eggs are for sale||She sells her eggs for $2 per egg and has 9 available for sale so 2 * 9 = $<<2*9=18>>18 per day||A: 18

- - - - - - - - - - - - - - - - - - - - - - - - - - - - - - - - - - - - - - - - - - - - - -

**DROP**

**Situation (Premise):** Hoping to rebound from their loss to the Patriots, the Raiders stayed at home for a Week 16 duel with the Houston Texans. Oakland would get the early lead in the first quarter as quarterback JaMarcus Russell completed a 20-yard touchdown pass to rookie wide receiver Chaz Schilens. The Texans would respond with fullback Vonta Leach getting a 1-yard touchdown run, yet the Raiders would answer with kicker Sebastian Janikowski getting a 33-yard and a 30-yard field goal. Houston would tie the game in the second quarter with kicker Kris Brown getting a 53-yard and a 24-yard field goal. Oakland would take the lead in the third quarter with wide receiver Johnnie Lee Higgins catching a 29-yard touchdown pass from Russell, followed up by an 80-yard punt return for a touchdown. The Texans tried to rally in the fourth quarter as Brown nailed a 40-yard field goal, yet the Raiders' defense would shut down any possible attempt. ***Who scored the first touchdown of the game?***

**Claim (Hypothesis):** Chaz Schilens

**Question:** Is the Claim supported by the Situation?

**Generated response:** Chaz Schilens was the receiver of the touchdown pass from JaMarcus Russell, so he scored the first touchdown. The answer is Yes.

Figure 1: Examples from the GSM8K and DROP datasets illustrating two explanation paradigms: Explain-then-Predict and Predict-then-Explain. For clarity, we use "||" to represent the Python newline character "\n".

In this work, we introduce a novel framework that harnesses the capabilities of LLMs to evaluate the model-generated reasoning chains, without the need of human-annotated reference chains. The application of the Socratic method has been demonstrated to enhance the quality of prompts for LLMs in reasoning tasks (Chang, 2023a; Dong et al., 2023). Building upon this insight, we further integrate the Socratic method to craft optimized prompts, facilitating better reference-free reasoning evaluations using LLMs, which we denote as SOCREVAL - representing the **Soc**ratic method for **R**easoning **Eval**uation. Specifically, we employ three fundamental strategies from the Socratic method—Definition, Maieutics, and Dialectic—and their combinations, aiming to refine the prompting mechanism of LLMs for reference-free reasoning evaluation.

To verify the efficacy of our proposed framework, LLMs with the Socratic method, we assessed its correlation with human judgment concerning the overall quality of reasoning chains produced by LLMs across four diverse datasets from ROSCOE (Golovneva et al., 2022): GSM8K (Cobbe et al., 2021) for arithmetic reasoning; e-SNLI (Camburu et al., 2018) for both deductive and commonsense reasoning; DROP (Dua et al., 2019) for discrete reasoning; and Cosmos QA (Huang et al., 2019) for commonsense reasoning. Our empirical findings reveal that GPT-4 exhibits a superior correlation with human judgment in comparison to existing reference-free reasoning evaluation metrics, notably ROSCOE (Golovneva et al., 2022) and RECEVAL (Prasad et al., 2023). By leveraging the Socratic method, SOCREVAL notably improves GPT-4's correlation coefficient with human judgment from $0.40$ to a remarkable $0.58$ when assessing the overall quality of the generated reasoning chains—surpassing even the performance of ROSCOE when furnished with human-written reasoning chains as references. A comprehensive analysis underscores the robustness of our proposed framework in terms of prompt writing and example selection while highlighting its cost-efficiency.

## 1.1 RELATED WORK

**Prompting LLMs with the Socratic method.** The Socratic method's essence is a sequence of probing questions to elucidate complex ideas, closely relevant to the LLM prompting techniques. Chang (2023a) crafted prompt templates utilizing the Socratic method, introducing the Critical Inquisitive Template (CRIT) for reasoning evaluation and subsequent Socratic synthesis for decision-making (Chang, 2023b). Dong et al. (2023) further employed the Socratic method for deeper LLM engagements in intricate problem-solving. Unlike these endeavors, which emphasize qualitative case analyses, our work focuses on quantitative experiments for reference-free reasoning evaluation.

**Evaluation of reasoning chains.** Evaluating the quality of reasoning chains generated by models has been traditionally approached by contrasting them with human-generated ones, referred to as reference-based reasoning evaluation. Conventional natural language generation (NLG) metrics assess the similarity between such machine-generated and human-crafted reasoning chains (Celikyilmaz et al., 2020; Clinciu et al., 2021; Welleck et al., 2022). In contrast, several domain-specific metrics were tailored for assessing reasoning chains relying on the specific structure of the dataset (Dalvi et al., 2021; Saparov & He, 2022; Han et al., 2022). Recently, ROSCOE (Golovneva et al., 2022) pioneered reference-free reasoning evaluation by introducing metrics grounded in step-by-step reasoning chains, targeting dimensions such as semantic consistency, logicality, informativeness, fluency, and factuality. RECEVAL (Prasad et al., 2023) moved further and centered its evaluation on correctness and informativeness. Both ROSCOE and RECEVAL align closely with our work's focus on reference-free reasoning evaluation. Notably, while both methods necessitate fine-tuning on datasets with human-derived reasoning chains, our approach avoids such fine-tuning requirements.

**Evaluating text generation with LLMs.** The stellar capabilities of LLMs on NLP tasks have propelled their adoption in evaluating generated text quality. Techniques range from harnessing conditional generative probabilities (Fu et al., 2023) to leveraging the prompts tailored for specific evaluation needs (Liu et al., 2023; Lu et al., 2023). Such methods have been deployed across diverse NLG domains, including summarization (Gao et al., 2023), machine translation (Kocmi & Federmann, 2023), and more (Wang et al., 2023), with evaluations being both individual and comparative (Chen et al., 2023; Zheng et al., 2023). In contrast to these endeavors, our research uses LLMs with the Socratic method to realize reference-free reasoning evaluation, whereas the aforementioned works target other text-generation tasks.

## 2 LARGE LANGUAGE MODELS WITH THE SOCRATIC METHOD

Our goal is to evaluate the overall quality of step-by-step reasoning without using references. This section elucidates our approach, wherein we harness the capabilities of LLMs, combined with the Socratic method, to undertake this assessment without human-crafted references.

### 2.1 PROMPT SKELETON FOR REFERENCE-FREE REASONING EVALUATION

Two dominant explanation paradigms often incorporate step-by-step reasoning chains to elucidate final answers: Explain-then-Predict (E-P) and Predict-then-Explain (P-E) (Ye & Durrett, 2022; Zelikman et al., 2022). Given the nature of LLMs such as GPT-4, it's crucial to formulate appropriate prompt templates to assess the reasoning chains within these paradigms.

**Explain-then-Predict (E-P).** Within this paradigm, the explanation precedes the final answer, both being part of the LLM's generated response. A concrete illustration is provided in Figure 1. We propose the subsequent prompt template for evaluating reasoning chains: ***Instruction** + Example question + Example generated response + Example representation + Question + Generated response + **Evaluation prompt***. The detailed prompt template can be found in Appendix A.1.

**Predict-then-Explain (P-E).** Here, the explanation follows the final answer. Notably, the explanation generated doesn't influence the final answer in P-E. For specific instances, refer to Figure 1 (see more in Figure 4 in Appendix A.1). In crafting prompts for this paradigm, we leverage terminology from the expert annotation user interface presented by Golovneva et al. (2022), including terms like Situation (Premise) and Claim (Hypothesis). Our proposed template is: ***Instruction** +*

*Example Situation (Premise) + Example Claim (Hypothesis) + Example question + Example generated response + Example representation + Situation (Premise) + Claim (Hypothesis) + Question + Generated response + **Evaluation prompt**.* Note that the question in this template is always "Is the Claim supported by the Situation?" which serves as a directive for LLMs to elucidate the provided Claim (Hypothesis) given the Situation (Premise). Beyond evaluating reasoning chains for question-answering explanations, this template is also suitable for assessing reasoning within natural language inference explanations. Detailed prompt templates are in Appendix A.1.

Despite the distinction between prompt templates for both paradigms, both employ ***the identical*** *Instruction* and *Evaluation prompt*. Drawing inspiration from Golovneva et al. (2022), the *Instruction* and *Evaluation prompt* used in our experiments are described below:

---

**(1) Instruction and Evaluation Prompt for GPT-4**:
Instruction: Does the generated response answer the question in a well-justified manner? Please give me an overall quality score in [1, 2, 3, 4, 5] (1=incomprehensible and wrong, 5=clear and correct).
(Evaluation prompt) Please give me the overall quality of the generated response for the question based on the instruction and the format of the example representation.

---

While the term "Instruction" is incorporated within the prompt, the term "Evaluation prompt" is omitted and is solely used to denote the final sentence of the prompt. Additionally, to ensure LLMs generate the intended output format, each of our prompt templates consistently integrates *one demonstration example* for reference-free reasoning evaluation.

## 2.2 REFINEMENT OF PROMPTS THROUGH THE SOCRATIC METHOD

The Socratic method, characterized by a series of probing questions aimed at exploring complex ideas, is a foundational approach in teaching and philosophy, fostering critical thinking and promoting self-discovery. Recent work has adopted this method to enhance prompts for LLMs, leading to enriched reasoning capacities (Chang, 2023a; Dong et al., 2023). Chang (2023a) identifies ten principal strategies within the Socratic method: Definition, Generalization, Induction, Elenchus, Hypothesis Elimination, Maieutics, Dialectic, Recollection, Irony, and Analogy. Similar to CRIT as decribed in (Chang, 2023a), we identify three strategies highly aligned with our use cases. Diverging from CRIT, we omit the Elenchus strategy from our framework. This decision stems from our observation that our datasets do not necessitate obtaining extra evidence beyond the provided context to support the reasoning chains. Nonetheless, we acknowledge its potential significance for reasoning tasks lacking adequate context, such as StrategyQA (Geva et al., 2021). Exploring this further is deferred to future work.

**Definition strategy.** Socrates frequently employed definitions to elucidate key terminologies. In reference-free reasoning evaluation, the definition strategy can be used to refine the comprehension of assessment criteria for LLMs. When incorporating this strategy into GPT-4, we denote the resultant evaluation metric as SOCREVAL (Definition). Differences, when compared with the original prompt (Prompt (1)), are highlighted in italicized text in purple.

---

**(2) Instruction and Evaluation Prompt for SOCREVAL (Definition)**:
Instruction: Does the generated response answer the question in a well-justified manner? Please give me an overall quality score in [1, 2, 3, 4, 5] (1=incomprehensible and wrong, 5=clear and correct). *Note that you need to take into account both the explanation and the answer in the generated response.*
(Evaluation prompt) Please give me the overall quality of the generated response for the question based on the instruction and the format of the example representation.

---

**Maieutics strategy.** Maieutics assists individuals in revealing their inherent knowledge. By applying maieutics, we prompt LLMs to analyze the quality of reasoning chains prior to delivering the final score. This bears similarity to chain-of-thought prompting (Wei et al., 2022; Kojima et al., 2022) with a divergent focus on reference-free reasoning evaluation. It's important to differentiate our approach from the Maieutic prompting in Jung et al. (2022). Although both are inspired by

the Maieutics strategy from the Socratic method, their methodology induces a tree of explanations through an abductive and recursive manner. With the integration of the maieutics strategy, the resultant evaluation metric is labeled SOCREVAL (Maieutics). Any deviations from the original prompt (Prompt (1)) are highlighted in italicized blue.

---

**(3) Instruction and Evaluation Prompt for SOCREVAL (Maieutics)**:
Instruction: Does the generated response answer the question in a well-justified manner? Please *conduct a qualitative analysis on the generated response first* and then give me an overall quality score in [1, 2, 3, 4, 5] (1=incomprehensible and wrong, 5=clear and correct) for the given generated response *by taking into account the qualitative analysis*.
(Evaluation prompt) Please *conduct a qualitative analysis on the generated response first* and then give me the overall quality of the given generated response for the question *by taking into account the qualitative analysis* based on the instruction and the format of the example representation:

---

**Dialectic strategy.** The dialectic approach navigates diverse perspectives through constructive discourse, fostering profound insights into the subject matter. Within the context of reference-free reasoning evaluation, we harness the dialectic strategy by prompting LLMs to formulate their own responses to a given question before evaluating existing reasoning chains. This methodology aligns with generating "pseudo references" in reference-free summarization evaluation, as introduced by (Gao et al., 2020; Chen et al., 2021). While their emphasis is on summarization assessment, ours is dedicated to reasoning evaluation. The resultant evaluation metric is termed SOCREVAL (Dialectic). Discrepancies against the original prompt (Prompt (1)) are highlighted in italicized brown.

---

**(4) Instruction and Evaluation Prompt for SOCREVAL (Dialectic)**:
Instruction: Does the generated response answer the question in a well-justified manner? Please *generate your own response for the question first* and then give me an overall quality score in [1, 2, 3, 4, 5] (1=incomprehensible and wrong, 5=clear and correct) for the given generated response *by taking into account your own response*.
(Evaluation prompt) Please *generate your own response for the question first* and then give me the overall quality of the given generated response for the question *by taking into account your own response* based on the instruction and the format of the example representation:

---

**Integration of three strategies.** We combine the aforementioned three strategies to devise optimized prompts for reference-free reasoning evaluation. The resultant evaluation metric is termed SOCREVAL (All)[1]. Further explorations of different combinations are detailed in Appendix A.1.

---

**(5) Instruction and Evaluation Prompt for SOCREVAL (All)**:
Instruction: Does the generated response answer the question in a well-justified manner? *Please generate your own response for the question first*, *then conduct a qualitative analysis on the generated response* *by taking into account your own response*, and finally give me an overall quality score in [1, 2, 3, 4, 5] (1=incomprehensible and wrong, 5=clear and correct) for the given generated response *by taking into account both* *your own response* and *the qualitative analysis*. *Note that you need to take into account both the explanation and the answer in the generated response.*
(Evaluation prompt) Please *generate your own response for the question first*, *then conduct a qualitative analysis on the generated response* *by taking into account your own response*, and finally give me the overall quality of the given generated response for the question *by taking into account both* *your own response* and *the qualitative analysis* based on the instruction and the format of the example representation:

---

[1]Unless otherwise specified, references to SOCREVAL implicitly denote SOCREVAL (All).

|  | GSM8K | e-SNLI | DROP | Cosmos QA | Average |
|---|---|---|---|---|---|
| Reference-based ROSCOE | 0.81 | 0.43 | - | - | - |
| Reference-free ROSCOE | 0.32 | 0.30 | 0.22 | 0.18 | 0.26 |
| RECEVAL | 0.36 | - | - | - | - |
| GPT-4 | 0.57 | 0.24 | 0.42 | 0.39 | 0.40 |
| SOCREVAL (Definition) | 0.62 | 0.30 | 0.43 | 0.39 | 0.44 |
| SOCREVAL (Maieutics) | 0.64 | **0.58** | 0.47 | 0.39 | 0.52 |
| SOCREVAL (Dialectic) | 0.74 | 0.31 | 0.43 | **0.43** | 0.48 |
| SOCREVAL (All) or SOCREVAL | **0.82** | **0.58** | **0.49** | 0.42 | **0.58** |

Table 1: Somers' D correlations between reasoning evaluation metrics and human judgment. For reference-free evaluation metrics (excluding reference-based ROSCOE), the top two correlations in each column are distinctly highlighted in **bold** and underlined. "-" indicates missing values: Prasad et al. (2023) assessed RECEVAL exclusively on GSM8K, and only GSM8K and e-SNLI encompass human-annotated explanations, enabling the acquisition of reference-based ROSCOE scores. For GPT-4 and SOCREVAL variants, all p-values of the correlations are below 0.0001 (****).

## 3 EXPERIMENTS

In this section, we evaluate the efficacy of our proposed framework, LLMs with the Socratic Method, in the context of reference-free reasoning evaluation.

**Datasets.** In our experiments, we utilize four[2] human judged datasets from ROSCOE (Golovneva et al., 2022), including GSM8K (Cobbe et al., 2021) for arithmetic reasoning, e-SNLI (Camburu et al., 2018) for deductive and commonsense reasoning, DROP (Dua et al., 2019) for discrete reasoning, and Cosmos QA(Huang et al., 2019) for commonsense reasoning. In the context of GSM8K, the reasoning chains in the ROSCOE datasets are derived from the GPT-3 175B Verification model (Cobbe et al., 2021), leveraging the chain-of-thought prompting approach (Wei et al., 2022). For the remaining datasets, GPT-3 (Brown et al., 2020) is used to extract detailed reasoning chains. Notably, GSM8K adopts the Explain-then-Predict explanatory paradigm, while the others are aligned with the Predict-then-Explain explanation paradigm. Expert annotators evaluated the reasoning chains for factors such as overall quality, and examined individual steps, identifying issues like commonsense errors. For the scope of this study, we only focus on the overall quality of reasoning chains, placing discussions on specific error types in Appendix A.2. The datasets comprise human-judged annotations on reasoning chains for 200 examples in GSM8K, 151 in ESNLI, 210 in DROP, and 195 in COSMOS-QA. We refer readers to Golovneva et al. (2022) for more details.

**Baselines.** We consider two established suites of reasoning evaluation metrics as our baselines: ROSCOE (Golovneva et al., 2022) and RECEVAL (Prasad et al., 2023). ROSCOE encompasses a comprehensive set of metrics, assessing attributes such as semantic consistency, logicality, informativeness, fluency, and factuality by exploiting the properties of step-by-step rationales. This suite incorporates both reference-free and reference-based metrics. In contrast, RECEVAL introduces a collection of purely reference-free metrics specifically crafted to measure the correctness and informativeness of reasoning chains. This is achieved by harnessing entailment and pointwise $\mathcal{V}$-information (Ethayarajh et al., 2022), leveraging granular claims in reasoning chains called reasoning content units. Considering the various metrics within reference-free ROSCOE, reference-based ROSCOE[3], and RECEVAL, we present **the peak performance** from each group of metrics as our baselines, though the optimal metric for each group often varies across different datasets.

**Meta evaluation.** In alignment with ROSCOE and RECEVAL, we employ Somers' D (Somers, 1962) to measure the correlation between human judgments and reasoning evaluation metrics.

---

[2]SemEVAL (Ostermann et al., 2018) was excluded as Golovneva et al. (2022) did not release their annotations for it due to approval constraints.

[3]While our focus is on reference-free reasoning evaluation, and thus we do not formally consider reference-based ROSCOE as our baseline, we include comparisons with them for a more comprehensive overview.

|  | GSM8K | e-SNLI | DROP | Cosmos QA | Average |
|---|---|---|---|---|---|
| SOCREVAL (All) or SOCREVAL | 0.82 | **0.58** | 0.49 | 0.42 | **0.58** |
| SOCREVAL (All - **Definition**) | **0.83** | 0.51 | 0.49 | 0.42 | 0.56 |
| SOCREVAL (All - **Maieutics**) | 0.81 | 0.38 | **0.51** | 0.42 | 0.53 |
| SOCREVAL (All - **Dialectic**) | 0.62 | 0.49 | 0.47 | **0.45** | 0.51 |

Table 2: Ablation study of SOCREVAL. For each column, the top two correlations are distinctly highlighted in **bold** and underlined. All reported correlations have p-values less than 0.0001 (****).

|  | GSM8K | e-SNLI | DROP | Cosmos QA | Average |
|---|---|---|---|---|---|
| SOCREVAL (6 different prompts) | $0.80 \pm 0.01$ | $0.56 \pm 0.04$ | $0.49 \pm 0.01$ | $0.42 \pm 0.03$ | $0.57 \pm 0.01$ |
| SOCREVAL (6 different examples) | $0.83 \pm 0.02$ | $0.54 \pm 0.03$ | $0.50 \pm 0.01$ | $0.42 \pm 0.02$ | $0.57 \pm 0.01$ |

Table 3: Assessment of SOCREVAL's robustness concerning prompt writing and example selection. The standard deviation of the performance is indicated post $\pm$ for each configuration.

Specifically, using Kendall's $\tau$ coefficient, Somers' D correlation is articulated as $D(Y|X) = \tau(X, Y)/\tau(X, X)$. Note that the Somers' D coefficient is asymmetric, necessitating that the human score is chosen as the first variable ($X$) and the metric score as the second variable ($Y$). Unless otherwise stated, we adopt practices from ROSCOE and RECEVAL to normalize scores to the $[0, 1]$ range for correlation analysis, even though Somers' D is inherently scale-invariant due to its focus on ordinal associations. A detailed analysis of the meta evaluation is in Section 4.

**Results.** As shown in Table 1, GPT-4 outperforms existing reference-free reasoning evaluation metrics, namely reference-free ROSCOE (on average) and RECEVAL. Note that while both reference-free ROSCOE and RECEVAL necessitate model fine-tuning on datasets furnished with human-annotated reasoning chains, GPT-4 operates effectively without such fine-tuning, underscoring its effectiveness in reference-free reasoning evaluation. By integrating three strategies derived from the Socratic method — namely, Definition, Maieutics, and Dialectic — our proposed evaluation metric, SOCREVAL, further outperforms GPT-4. Among these strategies, Maieutics emerges as the most effective on average. A fusion of the above three strategies not only augments GPT-4's performance but also surpasses that of reference-based ROSCOE. On average, these strategies from the Socratic method amplify GPT-4's correlation score from $0.40$ to $0.58$, highlighting the merit of the Socratic method in crafting suitable prompts for LLMs in reference-free reasoning evaluation.

## 4 ANALYSIS

To elucidate the efficacy of the Socratic method in crafting proper prompts for LLMs within the context of reference-free reasoning evaluation, we undertake an in-depth analysis of our proposed framework: LLMs with the Socratic method. For brevity and clarity, our primary emphasis centers on our best evaluation metric, SOCREVAL (All), often referred to simply as SOCREVAL.

**Ablation study.** The SOCREVAL approach harnesses three distinct strategies from the Socratic method—Definition, Maieutics, and Dialectic—to elicit reference-free reasoning evaluations from LLMs. To investigate the significance of each strategy within this integration of three strategies, we omit each from SOCREVAL and observe the consequences. Detailed prompts utilized in the ablation study are in Appendix A.1. As shown in Table 2, excluding any single strategy consistently decreases GPT-4's efficacy. Notably, the Dialectic strategy emerges as the pivotal one within the integration of three strategies, with its omission resulting in the most pronounced performance degradation. However, while Dialectic holds paramount importance within the integration of three strategies, Maieutics outperforms when strategies are incorporated individually (see Table 1). This suggests the complex interactions among the strategies. Delving into the intricate dynamics of how these strategies interplay in shaping LLM prompts is compelling and warranted for future exploration.

**Robustness analysis.** Prompting techniques, in certain contexts, have exhibited sensitivity to their specific phrasings (Kojima et al., 2022). To discern the influence of prompt writing on SOCREVAL,

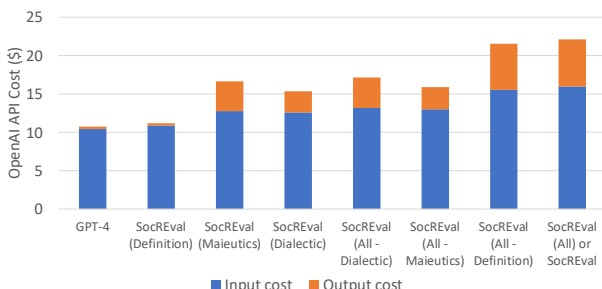

Figure 2: Cost comparison between GPT-4 and SOCREVAL. For each reasoning evaluation metric across four datasets, the OpenAI API costs are computed based on respective input (at $0.03 per $1K$ tokens) and output token counts (at $0.06 per $1K$ tokens).

|  | SD | $r$ | $\rho$ | $\tau$ | ACC | RMSE | MAE | Micro-F1 | Macro-F1 |
|---|---|---|---|---|---|---|---|---|---|
| GPT-4 | 0.40 | 0.55 | 0.55 | 0.49 | 0.48 | 1.40 | 0.91 | 0.48 | 0.28 |
| SOCREVAL | **0.58** | **0.69** | **0.66** | **0.58** | **0.52** | **1.07** | **0.68** | **0.52** | **0.39** |

Table 4: Comparison of SOCREVAL and GPT-4 across various meta-evaluation metrics. In addition to Somers' D (SD), metrics encompass Pearson's Correlation Coefficient ($r$), Spearman's Rank Correlation Coefficient ($\rho$), Kendall's $\tau$ ($\tau$), Accuracy (ACC), Root Mean Squared Error (RMSE), Mean Absolute Error (MAE), Micro F1 Score (Micro-F1), and Macro F1 Score (Macro-F1). It's crucial to note that for RMSE and MAE, lower values indicate superior performance, whereas for the remaining metrics, higher values are desirable. Note that scores for each configuration are computed as averages over four datasets: GSM8K, e-SNLI, DROP, and Cosmos QA.

we systematically rephrase Prompt (5) five times. More details are in Appendix A.1. By default, we utilize the first example from each dataset as the demonstration in the prompt. To systematically evaluate the influence of this demonstration example, we select five distinct examples at random for each dataset to serve as demonstrations. As shown in Table 3, the standard deviation for SOCREVAL across the six variations of Prompt (5) as well as the standard deviation encompassing six disparate demonstration examples both consistently measure at $0.01$, indicating the robustness of SOCREVAL on prompt writing and example selection.

**Cost analysis.** The substantial size of LLMs inherently implies considerable operating expenses that cannot be overlooked. Numerous methodologies have been devised to enhance LLMs' capabilities from varied dimensions, but they frequently come with escalated costs (Wei et al., 2022; Wang et al., 2022; Madaan et al., 2023). Within this context, we assess the OpenAI API costs tied to SOCREVAL in comparison to those of GPT-4. Note that these API costs are based on the number of input and output tokens processed by the LLMs. As illustrated in Figure 2, SOCREVAL (All) incurs a cost that is less than 2.1 times that of GPT-4, while amplifying the reasoning evaluation performance from $0.40$ to $0.58$. This highlights the cost-efficacy of our proposed framework.

**Meta evaluation analysis.** To obtain a deeper insight into the superiority of SOCREVAL over GPT-4, we expand our evaluation scope by incorporating eight supplementary metrics for meta evaluation beyond just Somers' D: Pearson's Correlation Coefficient, Spearman's Rank Correlation Coefficient, Kendall's $\tau$, Accuracy, Root Mean Squared Error (RMSE), Mean Absolute Error (MAE), Micro F1 Score, and Macro F1 Score. These meta-evaluation metrics are designed to quantify the alignment between evaluation metrics and human judgment. As shown in Table 4, SOCREVAL consistently outperforms GPT-4 across all meta-evaluation metrics. Furthermore, an examination of the confusion matrix (refer to Figure 5 in Appendix A.2) demonstrates a tendency in GPT-4 to overestimate the quality of reasoning chains. This overestimation is clearly alleviated in SOCREVAL, attributing to the incorporation of strategies from the Socratic method.

**Answer analysis.** When presented with a question, the whole reasoning trajectory comprises both an explanation and an answer, as shown in the E-P and P-E explanation paradigms. In this part, we

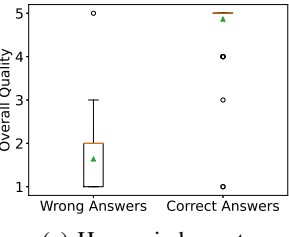 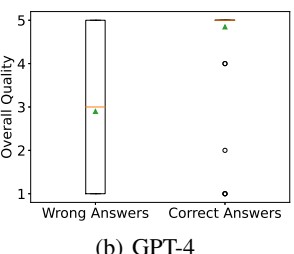 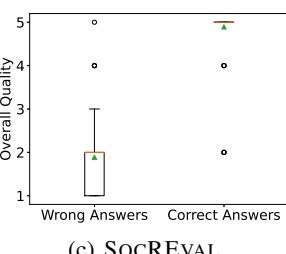

(a) Human judgment        (b) GPT-4        (c) SOCREVAL

Figure 3: Distribution of the quality of reasoning chains across questions on **GSM8K**, differentiated by answer correctness (correct versus wrong), as assessed through human judgment, GPT-4, and SOCREVAL. Quality scores span the discrete range [1, 2, 3, 4, 5], with the orange line representing the median and the green triangle signifying the mean.

delve deeper into the interplay between answer accuracy and the overall quality of reasoning chains. We segment our GSM8K examples into two groups: examples with correct answers and examples with wrong answers. Within each group, we investigate the distribution of the overall quality of reasoning chains, as assessed by human judgment, GPT-4, and SOCREVAL. Figure 3 reveals that, according to human evaluators, explanations associated with correct answers typically exhibit high quality, whereas those linked to wrong answers manifest a discernible decline in quality—a trend aligning with our anticipations. While GPT-4 demonstrates proficiency in assessing explanation quality for correctly answered instances, it tends to overestimate the quality for wrongly answered ones. By leveraging the Socratic method, SOCREVAL markedly mitigates this overestimation tendency. More analysis can be found in Figure 6 in Appendix A.2.

## 5 DISCUSSION

In this work, we introduced a novel approach that harnesses the Socratic method to craft optimized prompts for LLMs, specifically GPT-4, enabling enhanced assessment of reasoning chain quality. Distinctively, our framework avoids the need for model fine-tuning, a prevalent requirement in current reference-free reasoning evaluation metrics. The efficacy of our approach is substantiated across four datasets, wherein our proposed reference-free evaluation metric SOCREVAL demonstrates superior correlation with human judgments regarding reasoning chains produced by LLMs. Furthermore, comprehensive analyses underscore the robustness of our method with respect to prompt writing and example selection, all while maintaining cost-efficiency.

Similar to ROSCOE, we extended our analysis to evaluate the correlation between SOCREVAL and ten distinct human-annotated error types across the four datasets. As detailed in Table 12 within Appendix A.2, on average, SOCREVAL surpasses GPT-4 in almost all error types except the "Hallucination". Interestingly, when compared to reference-free ROSCOE, the SOCREVAL only lags in errors like "Hallucination," "Redundancy," and "Grammar." It is worthwhile to note that we prompt LLMs to assess the overall quality of reasoning chains rather than pinpointing specific error types. One prospective avenue might be directing LLMs to directly identify the presence of particular error types—a pursuit we postpone for future exploration. Additionally, while we employ GPT-4 to evaluate the quality of reasoning chains produced by GPT-3 variants, an intriguing inquiry emerges: Can GPT-4 effectively evaluate reasoning chains generated by itself? Addressing this necessitates human judgment of reasoning chains generated by GPT-4, a facet we postpone for future research.

In concluding this study, we anticipate several promising directions for future exploration. First, we simply select three strategies deeply relevant to our datasets; however, the expansive potential of the Socratic method's strategies deserves a more comprehensive exploration. Moreover, considering the important role of reasoning chains in advanced prompting techniques—such as chain of thought, tree of thoughts (Yao et al., 2023), and graph of thoughts (Besta et al., 2023), we plan to harness our framework for an enriched understanding of these techniques, subsequently designing better prompts. Since evaluation plays a pivotal role in feedback mechanisms, a fusion of our framework with prominent feedback integration methods like SELF-REFINE (Madaan et al., 2023) and Reflexion (Shinn et al., 2023) could be useful in improving the reasoning capabilities of LLMs.

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

## A  APPENDIX

In this section, we delve deeper into the experimental details and present additional results. Further information can be found in our submitted code.

### A.1  EXPERIMENTAL DETAILS

**Examples in e-SNLI and Cosmos QA.**  In addition to the examples from GSM8K and DROP depicted in Figure 1, we present examples from e-SNLI and Cosmos QA in Figure 4.

**Detailed prompting templates.**  For SOCREVAL, detailed prompting templates across the four datasets—GSM8K, e-SNLI, DROP, and Cosmos QA—are described in Tables 5-8. Detailed prompting templates for SOCREVAL with specific strategies, such as SOCREVAL (Definition) and SOCREVAL (All - Definition), can be readily derived by adjusting the *Instruction*, *Evaluation prompt*, and *Example representation* as appropriate.

**Prompting configurations in the ablation study.**  For the ablation study, the specific instructions and evaluation prompts employed with SOCREVAL (All - Definition), SOCREVAL (All - Maieutics), and SOCREVAL (All - Dialectic) are detailed in Prompts (6)-(8).

**Prompt variants in the robustness analysis.**  Building on the discussions in Section 4, for an in-depth robustness examination related to prompt writing for SOCREVAL, we have rephrased Prompt (5) five times. These five distinct variations of the original Prompt (5) are shown in Prompts (9)-(13).

### A.2  ADDITIONAL RESULTS

**Extended meta evaluation analysis.**  To better understand the advantages of SOCREVAL over GPT-4, we transitioned the initial five-class classification into both three-class and two-class classification. Within the three-class classification, the original score of $1$ is remapped to $-1$ (strongly negative), scores $2$ to $4$ are assigned to $0$ (moderately negative), and the score of $5$ is designated as $1$ (strongly positive). For the two-class classification, scores from $1$ to $4$ are remapped to $0$ (negative), while a score of $5$ corresponds to $1$ (positive). Similar to Section 4, meta evaluation is carried out for both two-class and three-class classification employing nine distinct metrics. As shown in Tables 9 and 10, it is evident that SOCREVAL systematically surpasses GPT-4 across all meta-evaluation metrics for both three-class and two-class classification. Delving deeper, we analyzed the confusion matrices for GPT-4 and SOCREVAL across the four datasets: GSM8K, e-SNLI, DROP, and Cosmos QA. Figure 5 reveals that while GPT-4 tends to overestimate the quality of reasoning chains, SOCREVAL successfully mitigates this bias by integrating strategies from the Socratic method.

**Extended answer analysis.**  Building upon the discussion in Section 4, we provide an in-depth visualization of the quality distributions for reasoning chains related to questions in the GSM8K dataset, as presented in Figure 6.

**Error type analysis.**  Drawing inspiration from ROSCOE and RECEVAL, we delve deeper into an analysis focused on specific error types[4] within reasoning chains. Initially, we compute the correlation between human-annotated errors and the overall quality of reasoning chains as judged by humans. As evident in Table 11, "Missing Step" and "Coherence" notably register the strongest correlations with the overall quality amongst the ten annotated error categories. This suggests these two errors heavily influence human assessment of reasoning chain quality. Additionally, we examine the correlation of both GPT-4 and SOCREVAL with ten human-annotated error types across the four datasets. As depicted in Table 12, SOCREVAL shows superior performance over GPT-4 in all error categories with the exception of "Hallucination". Furthermore, SOCREVAL outshines ROSCOE in all error types except "Hallucination", "Redundancy", and "Grammar". For a granular breakdown of correlations specific to each dataset, readers are directed to Tables 13-16.

---

[4]While some error types such as "Coherence" are evaluated at the chain-level, others like "Missing step" are assessed at the step-level. Consistent with the scoring setup in ROSCOE and RECEVAL, we aggregate step-level error scores using a "min" operation to derive the error score for the entire reasoning chain, based on the intuition that the quality of a reasoning chain is anchored by its weakest step.

---

**e-SNLI**

**Situation (Premise):** Two women are embracing while holding to go packages.
**Claim (Hypothesis):** Two woman are holding packages.
**Question:** Is the Claim supported by the Situation?
**Generated response:** The two women are most likely embracing because they are either friends or family. If they were just holding packages, there would be no need for them to embrace. The answer is Yes.

- - - - - - - - - - - - - - - - - - - - - - - - - - - - - - - - - - - - - - - - - - -

**Cosmos QA**

**Situation (Premise):** "Do i need to go for a legal divorce ? I wanted to marry a woman but she is not in the same religion , so i am not concern of the marriage inside church . I will do the marriage registered with the girl who i am going to get married . But legally will there be any complication , like if the other woman comes back one day , will the girl who i am going to get married now will be in trouble or Is there any complication ? ***Why is this person asking about divorce ?***
**Claim (Hypothesis):** He wants to get married to a different person .
**Question:** Is the Claim supported by the Situation?
**Generated response:** He wants to get married to a different person means he is not married to the person he wants to get married to. The answer is Yes.

---

Figure 4: Examples from the e-SNLI and Cosmos QA datasets.

---

**(6) Instruction and Evaluation Prompt for SOCREVAL (All - Definition):**

Instruction: Does the generated response answer the question in a well-justified manner? *Please generate your own response for the question first, then conduct a qualitative analysis on the generated response by taking into account your own response*, and finally give me an overall quality score in [1, 2, 3, 4, 5] (1=incomprehensible and wrong, 5=clear and correct) for the given generated response *by taking into account both your own response and the qualitative analysis*.

(Evaluation prompt) Please *generate your own response for the question first, then conduct a qualitative analysis on the generated response by taking into account your own response*, and finally give me the overall quality of the given generated response for the question *by taking into account both your own response and the qualitative analysis* based on the instruction and the format of the example representation:

---

**(7) Instruction and Evaluation Prompt for SOCREVAL (All - Maieutics):**

Instruction: Does the generated response answer the question in a well-justified manner? Please *generate your own response for the question first* and then give me an overall quality score in [1, 2, 3, 4, 5] (1=incomprehensible and wrong, 5=clear and correct) for the given generated response *by taking into account your own response. Note that you need to take into account both the explanation and the answer in the generated response.*

(Evaluation prompt) Please *generate your own response for the question first* and then give me the overall quality of the given generated response for the question *by taking into account your own response* based on the instruction and the format of the example representation:

---

**Instruction:** Does the generated response answer the question in a well-justified manner? Please generate your own response for the question first, then conduct a qualitative analysis on the generated response by taking into account your own response, and finally give me an overall quality score in [1, 2, 3, 4, 5] (1=incomprehensible and wrong, 5=clear and correct) for the given generated response by taking into account both your own response and the qualitative analysis. Note that you need to take into account both the explanation and the answer in the generated response.

**Example question:** Janet's ducks lay 16 eggs per day. She eats three for breakfast every morning and bakes muffins for her friends every day with four. She sells the remainder at the farmers' market daily for $2 per fresh duck egg. How much in dollars does she make every day at the farmers' market?

**Example generated response:** Janet eats 3 duck eggs for breakfast and bakes 4 into muffins so 3 + 4 = <<3+4=7>>7 duck eggs are used||Each day Janet's ducks lay 16 eggs and she uses 7, 16 - 7 = <<16-7=9>>9 duck eggs are for sale||She sells her eggs for $2 per egg and has 9 available for sale so 2 * 9 = $<<2*9=18>>18 per day||A: 18

**Example representation:** {"own response": "Janet uses a total of 3 (for breakfast) + 4 (for muffins) = 7 duck eggs each day. Given that her ducks lay 16 eggs per day, she is left with 16 - 7 = 9 eggs to sell at the farmers' market. Each of these eggs sells for $2, so Janet makes a total of 9 * $2 = $18 at the farmers' market each day.", "qualitative analysis": "The generated response is correct and complete. It precisely calculates the number of eggs that Janet consumes each day and subtracts that from the total number of eggs laid. It then multiplies the remaining eggs by the selling price to get the total amount that Janet makes each day. The response is also clear and easy to follow, with each step of the calculation laid out explicitly.", "overall quality": 5}

**Question:** A robe takes 2 bolts of blue fiber and half that much white fiber. How many bolts in total does it take?

**Generated response:** It takes 2 bolts of blue and half that much white so it takes 2*1/2 = <<2*1/2=1.0>>1 bolt of white||It takes 1 bolt of white and it takes 2 bolts of blue so it takes 1+2 = <<1+2=3>>3 bolts in total||A: 3

**(Evaluation prompt)** Please generate your own response for the question first, then conduct a qualitative analysis on the generated response by taking into account your own response, and finally give me the overall quality of the given generated response for the question by taking into account both your own response and the qualitative analysis based on the instruction and the format of the example representation:

Table 5: The full prompt for SOCREVAL on **GSM8K**. While elements like "Instruction" are embedded within the prompt, the term "Evaluation prompt" is exclusive and simply represents the last sentence in the prompt. For clarity, besides the demonstration example in the prompt, an additional example is provided. The notation "||" is employed to symbolize the Python newline character "\n".

---

**(8) Instruction and Evaluation Prompt for SOCREVAL (All - Dialectic):**

Instruction: Does the generated response answer the question in a well-justified manner? Please *conduct a qualitative analysis on the generated response first* and then give me an overall quality score in [1, 2, 3, 4, 5] (1=incomprehensible and wrong, 5=clear and correct) for the given generated response *by taking into account the qualitative analysis*. *Note that you need to take into account both the explanation and the answer in the generated response.*

(Evaluation prompt) Please *conduct a qualitative analysis on the generated response first* and then give me the overall quality of the given generated response for the question *by taking into account the qualitative analysis* based on the instruction and the format of the example representation:

**Instruction:** Does the generated response answer the question in a well-justified manner? Please generate your own response for the question first, then conduct a qualitative analysis on the generated response by taking into account your own response, and finally give me an overall quality score in [1, 2, 3, 4, 5] (1=incomprehensible and wrong, 5=clear and correct) for the given generated response by taking into account both your own response and the qualitative analysis. Note that you need to take into account both the explanation and the answer in the generated response.

**Example Situation (Premise):** Two women are embracing while holding to go packages.

**Example Claim (Hypothesis):** Two woman are holding packages.

**Example question:** Is the Claim supported by the Situation?

**Example generated response:** The two women are most likely embracing because they are either friends or family. If they were just holding packages, there would be no need for them to embrace. The answer is Yes.

**Example representation:** {"own response": "Yes, the Claim is supported by the Situation. The Situation states that two women are 'embracing while holding to go packages,' which means they are holding packages. Therefore, the Claim is accurate.", "qualitative analysis": "The generated response starts by making an assumption about why the two women might be embracing, suggesting they could be friends or family. While this could be true, it is not directly relevant to the question asked. The core of the question is about the Claim's accuracy given the Situation. The latter part of the generated response does correctly determine that the answer is 'Yes.' However, the earlier assumption makes the response longer and potentially more confusing.", "overall quality": 4}

**Situation (Premise):** Two women are embracing while holding to go packages.

**Claim (Hypothesis):** The men are fighting outside a deli.

**Question:** Is the Claim supported by the Situation?

**Generated response:** Two women are embracing while holding to go packages means they are friends or family. They are not fighting. The answer is No.

**(Evaluation prompt)** Please generate your own response for the question first, then conduct a qualitative analysis on the generated response by taking into account your own response, and finally give me the overall quality of the given generated response for the question by taking into account both your own response and the qualitative analysis based on the instruction and the format of the example representation:

Table 6: The full prompt for SOCREVAL on **e-SNLI**. While elements like "Instruction" are embedded within the prompt, the term "Evaluation prompt" is exclusive and simply represents the last sentence in the prompt. For clarity, besides the demonstration example in the prompt, an additional example is provided.

---

**(9) Instruction and Evaluation Prompt for SOCREVAL (Variant I):**

Instruction: Does the generated response aptly address the question in a well-reasoned way? First, create your own response to the question. Following this, conduct a qualitative analysis of the generated response, considering your own response. Conclude by providing an overall quality score ranging from [1, 2, 3, 4, 5] (1= incomprehensible and incorrect, 5= clear and accurate) for the given generated response, factoring in both your own response and the qualitative analysis. It's imperative to consider both the explanation and the answer present in the generated response.

(Evaluation prompt) Firstly, formulate your own response to the question. Then, undertake a qualitative analysis of the generated response, using your own response as a reference. Lastly, provide an assessment of the overall quality of the presented generated response by weighing both your own response and the qualitative analysis, ensuring adherence to the provided instruction and the format of the example representation:

**Instruction:** Does the generated response answer the question in a well-justified manner? Please generate your own response for the question first, then conduct a qualitative analysis on the generated response by taking into account your own response, and finally give me an overall quality score in [1, 2, 3, 4, 5] (1=incomprehensible and wrong, 5=clear and correct) for the given generated response by taking into account both your own response and the qualitative analysis. Note that you need to take into account both the explanation and the answer in the generated response.

**Example Situation (Premise):** Hoping to rebound from their loss to the Patriots, the Raiders stayed at home for a Week 16 duel with the Houston Texans. Oakland would get the early lead in the first quarter as quarterback JaMarcus Russell completed a 20-yard touchdown pass to rookie wide receiver Chaz Schilens. The Texans would respond with fullback Vonta Leach getting a 1-yard touchdown run, yet the Raiders would answer with kicker Sebastian Janikowski getting a 33-yard and a 30-yard field goal. Houston would tie the game in the second quarter with kicker Kris Brown getting a 53-yard and a 24-yard field goal. Oakland would take the lead in the third quarter with wide receiver Johnnie Lee Higgins catching a 29-yard touchdown pass from Russell, followed up by an 80-yard punt return for a touchdown. The Texans tried to rally in the fourth quarter as Brown nailed a 40-yard field goal, yet the Raiders' defense would shut down any possible attempt. ***Who scored the first touchdown of the game?***

**Example Claim (Hypothesis):** Chaz Schilens

**Example question:** Is the Claim supported by the Situation?

**Example generated response:** Chaz Schilens was the receiver of the touchdown pass from JaMarcus Russell, so he scored the first touchdown. The answer is Yes.

**Example representation:** {"own response": "Yes, the Claim is supported by the Situation as Chaz Schilens caught a 20-yard touchdown pass from JaMarcus Russell, making him the scorer of the first touchdown.", "qualitative analysis": "Both the generated response and my own response acknowledge that Chaz Schilens caught the first touchdown of the game from a pass by JaMarcus Russell. The generated response provides a concise and clear explanation that supports the answer.", "overall quality": 5}

**Situation (Premise):** Hoping to rebound from their loss to the Patriots, the Raiders stayed at home for a Week 16 duel with the Houston Texans. Oakland would get the early lead in the first quarter as quarterback JaMarcus Russell completed a 20-yard touchdown pass to rookie wide receiver Chaz Schilens. The Texans would respond with fullback Vonta Leach getting a 1-yard touchdown run, yet the Raiders would answer with kicker Sebastian Janikowski getting a 33-yard and a 30-yard field goal. Houston would tie the game in the second quarter with kicker Kris Brown getting a 53-yard and a 24-yard field goal. Oakland would take the lead in the third quarter with wide receiver Johnnie Lee Higgins catching a 29-yard touchdown pass from Russell, followed up by an 80-yard punt return for a touchdown. The Texans tried to rally in the fourth quarter as Brown nailed a 40-yard field goal, yet the Raiders' defense would shut down any possible attempt. ***How many field goals did Kris Brown kick?***

**Claim (Hypothesis):** 3

**Question:** Is the Claim supported by the Situation?

**Generated response:** Kris Brown kicked 4 field goals. The answer is No.

**(Evaluation prompt)** Please generate your own response for the question first, then conduct a qualitative analysis on the generated response by taking into account your own response, and finally give me the overall quality of the given generated response for the question by taking into account both your own response and the qualitative analysis based on the instruction and the format of the example representation:

Table 7: The full prompt for SOCREVAL on **DROP**. While elements like "Instruction" are embedded within the prompt, the term "Evaluation prompt" is exclusive and simply represents the last sentence in the prompt. For clarity, besides the demonstration example in the prompt, an additional example is provided.

**Instruction:** Does the generated response answer the question in a well-justified manner? Please generate your own response for the question first, then conduct a qualitative analysis on the generated response by taking into account your own response, and finally give me an overall quality score in [1, 2, 3, 4, 5] (1=incomprehensible and wrong, 5=clear and correct) for the given generated response by taking into account both your own response and the qualitative analysis. Note that you need to take into account both the explanation and the answer in the generated response.

**Example Situation (Premise):** Do i need to go for a legal divorce ? I wanted to marry a woman but she is not in the same religion , so i am not concern of the marriage inside church . I will do the marriage registered with the girl who i am going to get married . But legally will there be any complication , like if the other woman comes back one day , will the girl who i am going to get married now will be in trouble or Is there any complication ? ***Why is this person asking about divorce ?***

**Example Claim (Hypothesis):** He wants to get married to a different person .

**Example question:** Is the Claim supported by the Situation?

**Example generated response:** He wants to get married to a different person means he is not married to the person he wants to get married to. The answer is Yes.

**Example representation:** {"own response": "Yes, the Situation suggests that the person wants to get married to a different woman than the one he may have been previously involved with.", "qualitative analysis": "The generated response made an assumption that 'He wants to get married to a different person means he is not married to the person he wants to get married to.' While this may be true, it's not directly stated in the Situation. The Situation does not provide information about the current marital status of the person in question. However, the underlying premise that he wants to get married to a different person is accurate based on the context. The generated response does conclude correctly that the answer is 'Yes,' meaning the Claim is supported by the Situation.", "overall quality": 3}

**Situation (Premise):** Do i need to go for a legal divorce ? I wanted to marry a woman but she is not in the same religion , so i am not concern of the marriage inside church . I will do the marriage registered with the girl who i am going to get married . But legally will there be any complication , like if the other woman comes back one day , will the girl who i am going to get married now will be in trouble or Is there any complication ? ***Why is he worried about getting married ?***

**Claim (Hypothesis):** He was married before and she might come back one day .

**Question:** Is the Claim supported by the Situation?

**Generated response:** There might be a complication if the woman he was married to came back one day, due to the fact he is not married to her in a religious context, so she could come back and claim him as her husband. The answer is Yes.

**(Evaluation prompt)** Please generate your own response for the question first, then conduct a qualitative analysis on the generated response by taking into account your own response, and finally give me the overall quality of the given generated response for the question by taking into account both your own response and the qualitative analysis based on the instruction and the format of the example representation:

Table 8: The full prompt for SOCREVAL on **Cosmos QA**. While elements like "Instruction" are embedded within the prompt, the term "Evaluation prompt" is exclusive and simply represents the last sentence in the prompt. For clarity, besides the demonstration example in the prompt, an additional example is provided.

---

**(10) Instruction and Evaluation Prompt for SOCREVAL (Variant II):**

Instruction: Does the generated response aptly address the question with a well-substantiated justification? First, formulate your own response to the question. Subsequently, perform a qualitative analysis of the generated response, considering your own response. Finally, assign an overall quality score ranging from [1, 2, 3, 4, 5] (1= incomprehensible and incorrect, 5= lucid and accurate) for the given generated response. This score should reflect both your own response and the qualitative analysis. Ensure you consider both the explanation and the answer in the generated response.

(Evaluation prompt) First, craft your own response to the question. Next, undertake a qualitative analysis of the presented generated response, referencing your own response for context. Conclude by rating the overall quality of the generated response, integrating both your own response and the qualitative analysis, as guided by the instruction and the format of the example representation:

|  | SD | $r$ | $\rho$ | $\tau$ | ACC | RMSE | MAE | Micro-F1 | Macro-F1 |
|---|---|---|---|---|---|---|---|---|---|
| GPT-4 | 0.37 | 0.45 | 0.45 | 0.43 | 0.56 | 0.74 | 0.48 | 0.56 | 0.46 |
| SOCREVAL | **0.53** | **0.56** | **0.57** | **0.54** | **0.66** | **0.60** | **0.35** | **0.66** | **0.57** |

Table 9: Comparison between SOCREVAL and GPT-4 over diverse meta-evaluation metrics in the context of **three-class classification**. Specifically, we map the original score of 1 to $-1$ (strongly negative), scores 2 through 4 to 0 (moderately negative), and the original score of 5 to 1 (strongly positive). In addition to Somers' D (SD), metrics encompass Pearson's Correlation Coefficient ($r$), Spearman's Rank Correlation Coefficient ($\rho$), Kendall's $\tau$ ($\tau$), Accuracy (ACC), Root Mean Squared Error (RMSE), Mean Absolute Error (MAE), Micro F1 Score (Micro-F1), and Macro F1 Score (Macro-F1). It's crucial to note that for RMSE and MAE, lower values indicate superior performance, whereas for the remaining metrics, higher values are desirable. Scores for each configuration are computed as averages over four datasets: GSM8K, e-SNLI, DROP, and Cosmos QA.

---

**(11) Instruction and Evaluation Prompt for SOCREVAL (Variant III):**
Instruction: Does the generated response adequately address the question? First, formulate your own response to the question. Following this, perform a qualitative analysis of the generated response, using your own response as a reference. Conclude by assigning an overall quality score ranging from [1, 2, 3, 4, 5] (1 being incomprehensible and wrong, 5 being clear and correct) to the provided generated response. This score should consider both the clarity of the explanation and the accuracy of the answer in the generated response.
(Evaluation prompt) Begin by crafting your own response to the question. Next, undertake a qualitative analysis of the generated response, comparing it against your own response. Finally, rate the overall quality of the generated response, referencing both your own response and the qualitative analysis. Ensure your evaluation adheres to the given instruction and the format of the example representation:

---

**(12) Instruction and Evaluation Prompt for SOCREVAL (Variant IV):**
Instruction: Does the generated response adequately address the question with sound justification? First, provide your own response to the question. Subsequently, perform a qualitative analysis of the generated response, comparing it with your own response. Conclude by assigning an overall quality score ranging from [1, 2, 3, 4, 5] (1 = incomprehensible and wrong, 5 = clear and correct) to the generated response. This score should reflect both the clarity of the explanation and the accuracy of the answer in the generated response.
(Evaluation prompt) Begin by crafting your own response to the question. Then, execute a qualitative analysis of the generated response, using your own response as a benchmark. Finally, rate the overall quality of the provided generated response on a scale of [1 to 5], considering both your own response and the qualitative analysis, following the instruction provided and the format of the example representation:

---

**(13) Instruction and Evaluation Prompt for SOCREVAL (Variant V):**
Instruction: Does the generated response adequately address the question with valid justification? First, produce your own response to the question. Following this, conduct a qualitative analysis of the generated response, using your own response as a reference. Lastly, provide an overall quality score ranging from [1, 2, 3, 4, 5] (1=incomprehensible and wrong, 5=clear and correct) for the generated response. This score should consider both your own response and the qualitative analysis. It's important to assess both the explanation and the answer present in the generated response.
(Evaluation prompt) Begin by crafting your own response to the question. Then, perform a qualitative analysis of the generated response, referencing your own response. Conclude by assigning an overall quality rating to the provided generated response. This rating should factor in both your own response and the results of your qualitative analysis, following the provided instruction and the format of the example representation:

|  | SD | $r$ | $\rho$ | $\tau$ | ACC | RMSE | MAE | Micro-F1 | Macro-F1 |
|---|---|---|---|---|---|---|---|---|---|
| GPT-4 | 0.37 | 0.40 | 0.40 | 0.40 | 0.69 | 0.55 | 0.31 | 0.69 | 0.64 |
| SOCREVAL | **0.51** | **0.52** | **0.52** | **0.52** | **0.80** | **0.42** | **0.20** | **0.80** | **0.75** |

Table 10: Comparison between SOCREVAL and GPT-4 over diverse meta-evaluation metrics in the context of **two-class classification**. Specifically, original scores ranging from 1 to 4 are mapped to 0 (negative), while the original score of 5 is mapped to 1 (positive). In addition to Somers' D (SD), metrics encompass Pearson's Correlation Coefficient ($r$), Spearman's Rank Correlation Coefficient ($\rho$), Kendall's $\tau$ ($\tau$), Accuracy (ACC), Root Mean Squared Error (RMSE), Mean Absolute Error (MAE), Micro F1 Score (Micro-F1), and Macro F1 Score (Macro-F1). It's crucial to note that for RMSE and MAE, lower values indicate superior performance, whereas for the remaining metrics, higher values are desirable. Scores for each configuration are computed as averages over four datasets: GSM8K, e-SNLI, DROP, and Cosmos QA.

|  | GSM8K | e-SNLI | DROP | Cosmos QA | Average |
|---|---|---|---|---|---|
| COH | **0.73** | 0.15 | 0.33 | **0.56** | 0.44 |
| COMMON | 0.53 | 0.07 | 0.01 | 0.05 | 0.17 |
| FACT | 0.50 | 0.22 | 0.19 | 0.07 | 0.25 |
| HALL | 0.10 | 0.04 | 0.04 | -0.01 | 0.04 |
| RED | 0.10 | 0.14 | 0.13 | 0.04 | 0.10 |
| REP | 0.02 | - | 0.01 | 0.00 | - |
| LOGIC | 0.53 | 0.18 | 0.12 | 0.07 | 0.23 |
| MATH | 0.06 | 0.00 | 0.01 | - | - |
| GRAM | -0.01 | -0.01 | 0.02 | -0.02 | -0.01 |
| MISS | **0.73** | **0.36** | **0.50** | 0.35 | **0.49** |

Table 11: Somers' D correlations between human-annotated errors and human judgment of overall quality of reasoning chains. Evaluated errors encompass: Coherency (COH), Commonsense (COMMON), Factuality (FACT), Hallucination (HALL), Redundancy (RED), Repetition (REP), Logic (LOGIC), Arithmetic (MATH), Grammar (GRAM), and Missing Step (MISS). The highest and second-highest correlations for each column are indicated in **bold** and underlined, respectively. A '-' symbol indicates no correlation observed.

|  | QUAL | COH | COMMON | FACT | HALL | RED | REP | LOGIC | MATH | GRAM | MISS |
|---|---|---|---|---|---|---|---|---|---|---|---|
| Reference-free ROSCOE | 0.26 | 0.31 | **0.40** | 0.40 | **0.54** | **0.69** | - | 0.35 | - | **0.44** | 0.33 |
| RECEVAL | - | - | - | - | - | - | - | - | - | - | - |
| GPT-4 | 0.40 | 0.41 | 0.34 | 0.50 | 0.47 | 0.26 | - | 0.25 | - | -0.07 | 0.31 |
| SOCREVAL (Definition) | 0.44 | 0.44 | **0.40** | 0.46 | 0.43 | 0.25 | - | 0.28 | - | -0.04 | 0.40 |
| SOCREVAL (Maieutics) | 0.52 | 0.44 | 0.28 | 0.44 | 0.43 | 0.37 | - | 0.31 | - | -0.03 | 0.47 |
| SOCREVAL (Dialectic) | 0.48 | 0.42 | 0.35 | 0.54 | 0.38 | 0.32 | - | 0.33 | - | -0.06 | 0.37 |
| SOCREVAL (All - Dialectic) | 0.51 | 0.45 | 0.28 | 0.42 | 0.44 | 0.39 | - | 0.33 | - | 0.06 | 0.51 |
| SOCREVAL (All - Maieutics) | 0.53 | 0.48 | 0.34 | 0.53 | 0.48 | 0.39 | - | 0.36 | - | -0.06 | 0.47 |
| SOCREVAL (All - Definition) | 0.56 | 0.47 | 0.34 | **0.57** | 0.41 | 0.36 | - | **0.42** | - | 0.02 | 0.46 |
| SOCREVAL (All) or SOCREVAL | **0.58** | **0.51** | **0.40** | 0.53 | 0.43 | 0.34 | - | 0.36 | - | -0.03 | **0.52** |

Table 12: Average correlation analysis, computed across four datasets (GSM8K, e-SNLI, DROP, and Cosmos QA), between reasoning evaluation metrics and various aspects of human judgment. Beyond the overall quality (QUAL), ten specific error types are considered: Coherency (COH), Commonsense (COMMON), Factuality (FACT), Hallucination (HALL), Redundancy (RED), Repetition (REP), Logic (LOGIC), Arithmetic (MATH), Grammar (GRAM), and Missing Step (MISS). For each column, the highest and second-highest correlations are denoted in **bold** and underlined, respectively. The '-' symbol signifies an absence of observed correlation.

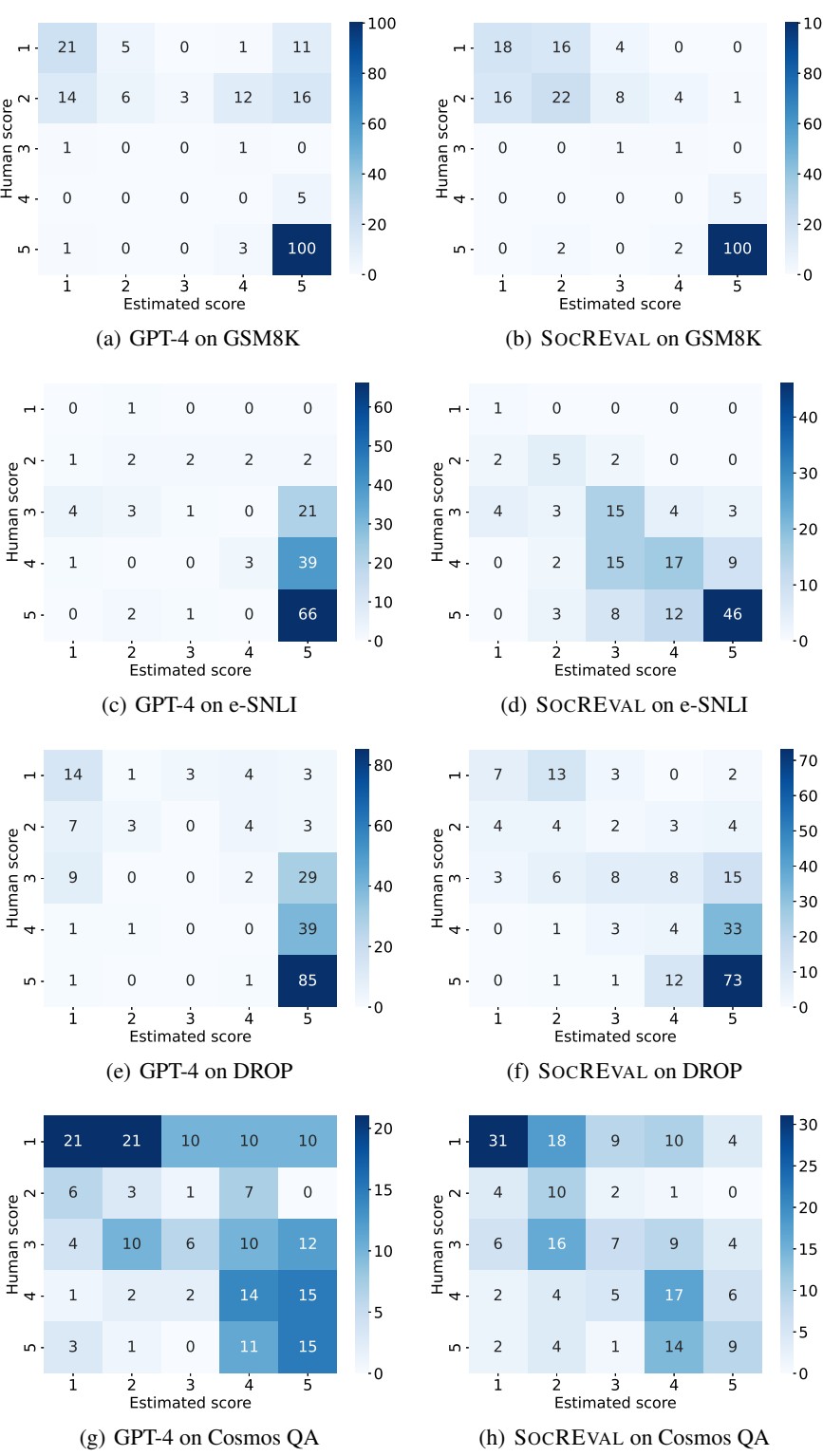

Figure 5: Confusion matrices for GPT-4 and SOCREVAL evaluated across four datasets: GSM8K, e-SNLI, DROP, and Cosmos QA.

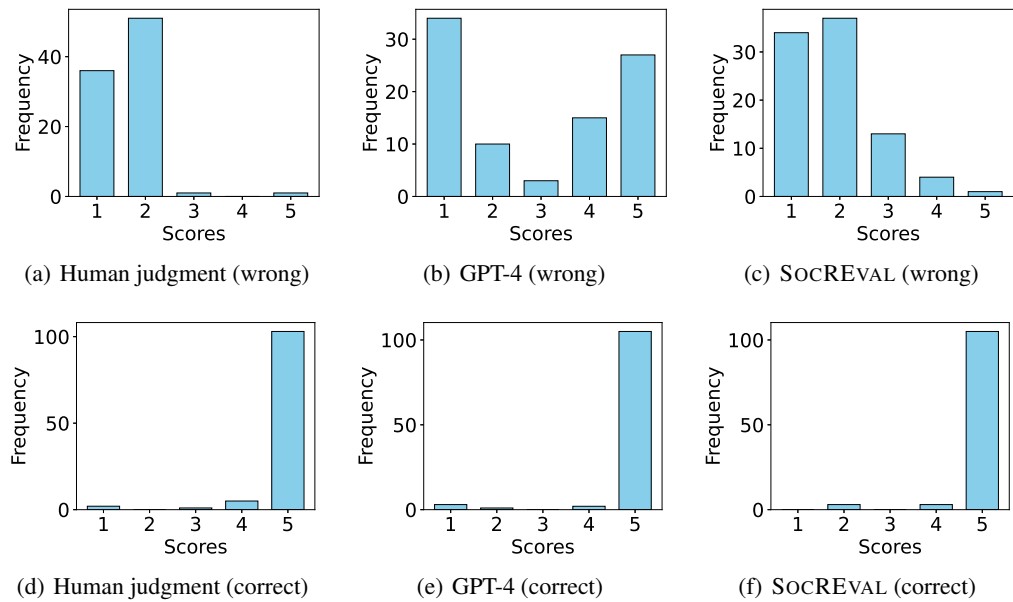

Figure 6: Detailed distributions of the quality of reasoning chains across questions on **GSM8K**, differentiated by answer correctness (correct versus wrong), as assessed through human judgment, GPT-4, and SOCREVAL.

|  | QUAL | COH | COMMON | FACT | HALL | RED | REP | LOGIC | MATH | GRAM | MISS |
|---|---|---|---|---|---|---|---|---|---|---|---|
| Reference-free ROSCOE | 0.32 | 0.30 | 0.28 | 0.35 | 0.50 | **0.64** | **0.98** | 0.22 | **0.56** | **0.34** | 0.35 |
| RECEVAL | 0.36 | 0.31 | 0.21 | 0.37 | 0.28 | 0.55 | 0.87 | 0.25 | 0.32 | - | - |
| GPT-4 | 0.57 | 0.47 | 0.51 | 0.51 | 0.61 | 0.33 | 0.82 | 0.54 | 0.27 | -0.16 | 0.66 |
| SOCREVAL (Definition) | 0.62 | 0.53 | 0.60 | 0.57 | **0.70** | 0.33 | 0.71 | 0.65 | 0.22 | -0.17 | 0.74 |
| SOCREVAL (Maieutics) | 0.64 | 0.54 | 0.67 | 0.54 | 0.63 | 0.41 | 0.77 | 0.66 | 0.31 | -0.02 | 0.75 |
| SOCREVAL (Dialectic) | 0.74 | 0.63 | 0.71 | 0.65 | 0.44 | 0.45 | 0.78 | 0.76 | 0.45 | -0.30 | 0.86 |
| SOCREVAL (All - Dialectic) | 0.62 | 0.52 | 0.64 | 0.56 | 0.55 | 0.41 | 0.82 | 0.64 | 0.39 | -0.00 | 0.75 |
| SOCREVAL (All - Maieutics) | 0.81 | 0.67 | 0.78 | **0.72** | 0.69 | 0.47 | 0.80 | **0.80** | 0.44 | -0.32 | **0.92** |
| SOCREVAL (All - Definition) | **0.83** | 0.68 | **0.79** | 0.70 | 0.60 | 0.46 | 0.68 | 0.78 | 0.46 | -0.31 | 0.91 |
| SOCREVAL (All) or SOCREVAL | 0.82 | **0.72** | 0.78 | 0.71 | 0.63 | 0.44 | 0.84 | 0.76 | 0.50 | -0.24 | **0.92** |

Table 13: Correlation analysis between reasoning evaluation metrics and diverse facets of human judgment on **GSM8K**. Beyond the overall quality (QUAL), ten specific error types are also considered: Coherency (COH), Commonsense (COMMON), Factuality (FACT), Hallucination (HALL), Redundancy (RED), Repetition (REP), Logic (LOGIC), Arithmetic (MATH), Grammar (GRAM), and Missing Step (MISS). For each column, the highest and second-highest correlations are denoted in **bold** and underlined, respectively. The '-' symbol signifies an absence of observed correlation.

|  | QUAL | COH | COMMON | FACT | HALL | RED | REP | LOGIC | MATH | GRAM | MISS |
|---|---|---|---|---|---|---|---|---|---|---|---|
| Reference-free ROSCOE | 0.30 | 0.35 | 0.40 | 0.43 | **0.80** | **0.62** | - | 0.53 | **0.60** | **0.41** | **0.60** |
| RECEVAL | - | - | - | - | - | - | - | - | - | - | - |
| GPT-4 | 0.24 | 0.42 | **0.43** | 0.65 | 0.33 | 0.07 | - | 0.29 | -0.15 | -0.16 | -0.04 |
| SOCREVAL (Definition) | 0.30 | 0.44 | 0.38 | 0.61 | 0.29 | 0.08 | - | 0.23 | -0.21 | -0.05 | 0.17 |
| SOCREVAL (Maieutics) | **0.58** | 0.34 | 0.07 | 0.55 | 0.24 | 0.35 | - | 0.43 | 0.27 | -0.25 | 0.45 |
| SOCREVAL (Dialectic) | 0.31 | 0.33 | 0.24 | 0.70 | 0.32 | 0.15 | - | 0.39 | -0.21 | -0.04 | -0.04 |
| SOCREVAL (All - Dialectic) | 0.49 | 0.42 | 0.06 | 0.36 | 0.35 | 0.34 | - | 0.47 | -0.47 | -0.03 | 0.50 |
| SOCREVAL (All - Maieutics) | 0.38 | 0.39 | 0.14 | 0.55 | 0.44 | 0.31 | - | 0.44 | -0.34 | -0.35 | 0.22 |
| SOCREVAL (All - Definition) | 0.51 | 0.39 | 0.18 | **0.75** | 0.35 | 0.42 | - | **0.56** | 0.56 | 0.12 | 0.16 |
| SOCREVAL (All) or SOCREVAL | **0.58** | **0.49** | 0.34 | 0.65 | 0.46 | 0.26 | - | 0.47 | 0.00 | -0.05 | 0.41 |

Table 14: Correlation analysis between reasoning evaluation metrics and diverse facets of human judgment on **e-SNLI**. Beyond the overall quality (QUAL), ten specific error types are also considered: Coherency (COH), Commonsense (COMMON), Factuality (FACT), Hallucination (HALL), Redundancy (RED), Repetition (REP), Logic (LOGIC), Arithmetic (MATH), Grammar (GRAM), and Missing Step (MISS). For each column, the highest and second-highest correlations are denoted in **bold** and underlined, respectively. The '-' symbol signifies an absence of observed correlation.

| | QUAL | COH | COMMON | FACT | HALL | RED | REP | LOGIC | MATH | GRAM | MISS |
|---|---|---|---|---|---|---|---|---|---|---|---|
| Reference-free ROSCOE | 0.22 | 0.40 | **0.64** | 0.51 | 0.54 | **0.82** | **0.95** | **0.29** | **0.85** | **0.45** | 0.21 |
| RECEVAL | - | - | - | - | - | - | - | - | - | - | - |
| GPT-4 | 0.42 | 0.44 | 0.49 | 0.55 | 0.54 | 0.27 | -0.25 | 0.22 | 0.16 | 0.14 | 0.33 |
| SocREval (Definition) | 0.43 | 0.45 | 0.51 | 0.48 | 0.48 | 0.25 | -0.27 | 0.20 | 0.14 | 0.13 | 0.35 |
| SocREval (Maieutics) | 0.47 | 0.53 | 0.40 | 0.47 | 0.50 | 0.40 | 0.61 | 0.13 | 0.22 | 0.23 | 0.37 |
| SocREval (Dialectic) | 0.43 | 0.40 | 0.41 | **0.64** | **0.58** | 0.28 | -0.28 | 0.12 | 0.29 | 0.30 | 0.31 |
| SocREval (All - Dialectic) | 0.47 | 0.46 | 0.42 | 0.53 | 0.43 | 0.39 | 0.44 | 0.18 | 0.24 | 0.31 | 0.39 |
| SocREval (All - Maieutics) | **0.51** | **0.56** | 0.39 | 0.60 | 0.57 | 0.40 | 0.50 | 0.14 | 0.26 | 0.42 | 0.36 |
| SocREval (All - Definition) | 0.49 | 0.47 | 0.39 | 0.61 | 0.53 | 0.33 | 0.02 | 0.22 | 0.29 | 0.28 | 0.39 |
| SocREval (All) or SocREval | 0.49 | 0.51 | 0.43 | 0.56 | 0.47 | 0.36 | 0.53 | 0.18 | 0.18 | 0.38 | **0.42** |

Table 15: Correlation analysis between reasoning evaluation metrics and diverse facets of human judgment on **DROP**. Beyond the overall quality (QUAL), ten specific error types are also considered: Coherency (COH), Commonsense (COMMON), Factuality (FACT), Hallucination (HALL), Redundancy (RED), Repetition (REP), Logic (LOGIC), Arithmetic (MATH), Grammar (GRAM), and Missing Step (MISS). For each column, the highest and second-highest correlations are denoted in **bold** and underlined, respectively. The '-' symbol signifies an absence of observed correlation.

| | QUAL | COH | COMMON | FACT | HALL | RED | REP | LOGIC | MATH | GRAM | MISS |
|---|---|---|---|---|---|---|---|---|---|---|---|
| Reference-free ROSCOE | 0.18 | 0.19 | **0.28** | **0.29** | 0.32 | **0.66** | **0.67** | **0.35** | - | **0.56** | 0.15 |
| RECEVAL | - | - | - | - | - | - | - | - | - | - | - |
| GPT-4 | 0.39 | 0.30 | -0.06 | **0.29** | 0.39 | 0.39 | -0.34 | -0.04 | - | -0.09 | 0.28 |
| SocREval (Definition) | 0.39 | 0.35 | 0.10 | 0.19 | 0.24 | 0.36 | -0.36 | 0.05 | - | -0.05 | 0.33 |
| SocREval (Maieutics) | 0.39 | 0.34 | -0.02 | 0.19 | 0.36 | 0.31 | -0.28 | 0.03 | - | -0.06 | 0.30 |
| SocREval (Dialectic) | 0.43 | 0.32 | 0.05 | 0.19 | 0.18 | 0.39 | -0.32 | 0.05 | - | -0.20 | 0.35 |
| SocREval (All - Dialectic) | **0.45** | **0.40** | -0.00 | 0.24 | **0.43** | 0.41 | -0.40 | 0.03 | - | -0.05 | **0.39** |
| SocREval (All - Maieutics) | 0.42 | 0.33 | 0.04 | 0.25 | 0.22 | 0.38 | -0.18 | 0.08 | - | 0.01 | 0.36 |
| SocREval (All - Definition) | 0.42 | 0.34 | 0.00 | 0.22 | 0.15 | 0.23 | -0.13 | 0.11 | - | -0.02 | 0.36 |
| SocREval (All) or SocREval | 0.42 | 0.32 | 0.06 | 0.21 | 0.14 | 0.30 | -0.38 | 0.03 | - | -0.20 | 0.33 |

Table 16: Correlation analysis between reasoning evaluation metrics and diverse facets of human judgment on **Cosmos QA**. Beyond the overall quality (QUAL), ten specific error types are considered: Coherency (COH), Commonsense (COMMON), Factuality (FACT), Hallucination (HALL), Redundancy (RED), Repetition (REP), Logic (LOGIC), Arithmetic (MATH), Grammar (GRAM), and Missing Step (MISS). For each column, the highest and second-highest correlations are denoted in **bold** and underlined, respectively. The '-' symbol signifies an absence of observed correlation.

