# OpenReview forum: "SocREval: LLMs with the Socratic Method for Reference-free Reasoning Evaluation"
_ICLR.cc/2024/Conference — ICLR 2024 Conference Withdrawn Submission_

### Official Review · Reviewer_Hwhe · 2023-10-31

**Soundness:** 3 good
**Presentation:** 3 good
**Contribution:** 2 fair
**Rating:** 5
**Confidence:** 4

**Summary:**

The paper introduces a new method, SOCREVAL (Socratic method for Reasoning Evaluation), to evaluate the quality of reasoning chains produced by models without the need for human-annotated reference chains. The Socratic method, characterized by probing questions to clarify intricate ideas, is employed to create optimized prompts for Large Language Models (LLMs) to perform reference-free reasoning evaluation. The empirical results on four diverse datasets showed that this method significantly improved GPT-4’s performance in assessing the quality of reasoning chains, surpassing other reference-free and reference-based metrics.

**Strengths:**

1. The paper is well-written and well-structured. It studies a valuable research problem - evaluating the reasoning chain of models in an automatic manner without relying on human-crafted references. The need for scalable, efficient, and reference-free reasoning evaluation metrics in the context of large language models is evident, and the paper is dedicated to addressing this gap.

2. The paper proposes a simple approach SOCREVAL, which leverages the Socratic method-enhanced GPT-4 to provide reference-free reasoning evaluations. By employing strategies like Definition, Maieutics, and Dialectic from the Socratic method, this approach refines the prompting mechanism of LLMs, offering an effective way to evaluate reasoning chains without necessitating human-annotated references.

3. The paper shows the effectiveness of SOCREVAL through empirical tests on four human-annotated datasets. These experiments revealed that GPT-4, when integrated with SOCREVAL, outperforms existing reference-free and reference-based reasoning evaluation metrics. Also, the authors provide evidence that SOCREVAL improves the evaluation ability of GPT-4 on multiple metrics.

**Weaknesses:**

1. While the incorporation of the Socratic method is effective, its application in prompt optimization seems to lack a certain novelty. The authors themselves reference other works that have employed the Socratic method for enhancing LLM prompting techniques. Thus, the differentiation of this research from the existing literature needs to be articulated more convincingly.

2. My main concern revolves around the claims and experiment settings in the paper.
1）The paper claims that the proposed framework is "both cost-efficient and robust to prompt writing and example selection" needs stronger experimental validation. Although the cost analysis demonstrates that SOCREVAL incurs a cost that is less than 2.1 times that of GPT-4, the previous method, ROSCOE, uses a small language model that is far less costly than GPT-4. Additionally, the prompt robustness experiment lacks an exploration of the effect of different numbers of demonstrations on the performance, e.g., the number of demonstrations is 0.
2) The paper does not compare SOCREVAL with other smaller LMs besides GPT-4, such as T5 or LLAMA. It is unclear how SOCREVAL would perform with different LLMs as evaluators.
3) The paper does not provide any qualitative examples or case studies of SOCREVAL’s outputs or errors. It would be helpful to see some concrete examples of how SOCREVAL generates prompts, scores reasoning chains, and explains its decisions.

**Questions:**

Some questions that I have after reading the paper are:
1. Is it possible that the GPT-4 has learned from these datasets leading to biased assessment results? Could you prove that the proposed method has good generalisation on unseen datasets?
2. Is there a difference between multiple assessments of the GPT-4 for the same instance?
3. Given that GPT-4 is likely to be much larger than the models in the previous method (ROSCOE), is it fair to compare them?

---

### Official Review · Reviewer_TsV2 · 2023-11-01

**Soundness:** 2 fair
**Presentation:** 3 good
**Contribution:** 2 fair
**Rating:** 3
**Confidence:** 4

**Summary:**

The authors argue that existing reference-based approaches are labor intensive and are not necessarily unique, making comparison difficult. On the other hand, reference-free approaches require further fine-tuning on human annotation references, making the evaluation costly and not generalizable. The authors propose a socratic method for reference-free evaluation of LLM reasoning. They pick 3 different strategies to improve prompting performance: definition, maieutics, and dialectic. Each of these strategies slightly modify a prompt with additional requirements and constraints. Empirical results suggest that the combination of all strategies outperform baseline GPT-4 and each one is important to achieve the best results. The method is also shown to be robust to different wordings of prompts.

**Strengths:**

Socratic method is a general approach to improving and devising prompts that could generally be applied to any task. Empirical results suggest that even GPT-4 performance can be improved via minor but focused improvements to prompting.

**Weaknesses:**

The novelty of the paper compared to previous approaches that use Socratic method is not very clear. There are a few parts that need clarification.

1. Can you please highlight how your approach to guiding LLM prompts via socratic method is different than previous work? You explain one difference which is emphasizing more quantitative results but it is not clear if your approach results in a completely different way of prompting.

2. Related to the above, can some of the prompt statements from CRIT transfer to your problem? Do I need a different prompt statement if I want to use Socratic method for different tasks?

3. You mention that your method is cost efficient but I couldn’t find any reference point. The cost more than doubles while the performance improves 0.18 percent. Is 2.1/0.18 a good ratio to call a method cost efficient?

4. Would your approach work for other models other than GPT-4 such as GPT-3.5, Llama-2? Is there anything special about GPT-4 that makes it a better with for socratic method?

5. Why for the definition strategy, evaluation prompt is unchanged while for other strategies both evaluation and instructions prompts change?

**Questions:**

1. What is the main novelty of your work compared to other works that use socratic method in addition to more quantitative results?

2. Can prompt statements from CRIT or others that use socratic method transfer to your domain?

3. What makes a prompting method cost efficient?

4. Does your approach generalize to other LLMs?

---

### Official Review · Reviewer_QzWF · 2023-11-01

**Soundness:** 3 good
**Presentation:** 3 good
**Contribution:** 2 fair
**Rating:** 5
**Confidence:** 4

**Summary:**

The paper introduces SocREval, a novel framework that utilizes GPT-4 and the Socratic method for scalable and efficient reference-free evaluation of reasoning chains produced by large language models (LLMs). The framework addresses the limitations of existing evaluation methods by eliminating the need for labor-intensive human-annotated reasoning chains and circumventing the generalizability issues associated with reference-free metrics that require fine-tuning. Empirical evaluation across diverse datasets demonstrates that SocREval significantly outperforms existing reasoning evaluation metrics, showcasing a higher correlation with human judgment and proving to be both cost-efficient and robust to various prompts and examples. The integration of the Socratic method into the evaluation process stands out as a key innovation, enhancing the quality and reliability of reasoning assessments.

**Strengths:**

- Leveraging the strong reasoning ability of GPT-4, the paper designs a reference-free reasoning evaluation method to automatically evaluate reasoning chain quality.

- Leveraging the Socratic method significantly enhances the quality of reasoning evaluation, ensuring a higher correlation with human judgment and establishing the framework’s reliability in assessing reasoning capabilities.

- SocREval addresses the need for scalable and efficient reasoning evaluation methods, eliminating the dependency on labor-intensive human-annotated reasoning chains and offering a practical solution for large-scale applications.

**Weaknesses:**

- The framework’s reliance on GPT-4’s strong reasoning ability and robustness, raises concerns about its adaptability to other LLMs. A more extensive validation across different models (some smaller or weaker models like GPT-3.5/Claude/Llama2) would enhance the generalizability of the findings.
- Different sub-types of SOCREVAL metrics have different performances on the different tasks, so it may cost much more resources to do prompt engineering to find the optimal evaluation template. A more exhaustive analysis exploring a wider range of prompt variations would help in establishing the framework’s resilience to different input scenarios.
- While the paper claims cost-efficiency as a strength, a more detailed breakdown of the cost implications, compared to other existing methods, would provide tangible evidence to support this claim.
- The paper critiques existing methods for their reliance on human-crafted reasoning chains, which may not be unique. However, it does not explicitly address how SocREval overcomes this challenge. Providing clarity on this aspect could strengthen the argument for SocREval’s novelty and effectiveness.

**Questions:**

- Given GPT-4’s continuous updates, how can SocREval ensure a stable and fair comparison basis for future research on different methods?

- What guidelines does SocREval provide for choosing the most appropriate Socratic strategy (Definition, Maieutics, Dialectic) for new datasets?

- How does SocREval address and mitigate potential biases in its human-annotated validation datasets?

---

### Official Review · Reviewer_fFgo · 2023-11-07

**Soundness:** 3 good
**Presentation:** 3 good
**Contribution:** 2 fair
**Rating:** 3
**Confidence:** 4

**Summary:**

This paper proposes a simple prompting method for evaluating reasoning chains in the reference-free setting. The authors manually devise prompts using three strategies including definition, maieutics, and dialectic. Experimental results show that the proposed prompting method can further enhance GPT-4’s performance in the task of reasoning evaluation.

**Strengths:**

1. The proposed prompting method is simple and direct, which shows good empirical results.
2. This paper is overall easy to follow.

**Weaknesses:**

1. The novelty of the proposed method is limited. The authors use the term “socratic method” in the title and main content for multiple times. However, in my view, the proposed three strategies have been mostly studied in the existing works [1, 2, 3] possibly with different names. The authors only adapt these techniques to a different evaluation task (i.e., reasoning evaluation) without task-specific design, which provide few insights into the prompt design principle for an emerging evaluation task.

2. From the experiment, the proposed prompting method is only validated on GPT-4. I wonder whether this method can improve the evaluation performance of other base models such as ChatGPT and open-sourced LLMs (e.g., LLaMA). If the proposed method only works well on GPT-4, its contribution will be relatively thin and GPT-4’s ability may become the dominating factor for success.

3. From Table 13-16, SocREval still underperforms ROSCOE in most of the error types on all the datasets. This needs more analysis because the ability to detect various types of reasoning errors is as important as providing an overall quality score in the reasoning evaluation task. Existing works like ROSCOE and RECEVAL mostly report the results on different reasoning error types in the main content.

4. From Section 2.2, the authors instruct GPT-4 to generate qualitative analysis and pseudo references to help obtain a more accurate overall quality score. The quality of generated qualitative analysis and pseudo references should be assessed via human evaluation.


[1] G-EVAL: Nlg evaluation using gpt-4 with better human alignment.

[2] A training-free and reference-free summarization evaluation metric via centrality-weighted relevance and self-referenced redundancy. ACL 2021.

[3] Judging LLM-as-a-Judge with MT-Bench and Chatbot Arena.

**Questions:**

I have included my questions in the weaknesses part.